# N-terminal VP1 Truncations Favor *T* = 1 Norovirus-Like Particles

**DOI:** 10.3390/vaccines9010008

**Published:** 2020-12-24

**Authors:** Ronja Pogan, Victor U. Weiss, Kevin Bond, Jasmin Dülfer, Christoph Krisp, Nicholas Lyktey, Jürgen Müller-Guhl, Samuele Zoratto, Günter Allmaier, Martin F. Jarrold, Cesar Muñoz-Fontela, Hartmut Schlüter, Charlotte Uetrecht

**Affiliations:** 1Heinrich Pette Institute, Leibniz Institute for Experimental Virology, 20251 Hamburg, Germany; ronja.pogan@leibniz-hpi.de (R.P.); jasmin.duelfer@leibniz-hpi.de (J.D.); juergen.mueller@leibniz-hpi.de (J.M.-G.); 2European XFEL GmbH, 22869 Schenefeld, Germany; 3Institute of Chemical Technologies and Analytics, TU Wien, 1060 Vienna, Austria; victor.weiss@tuwien.ac.at (V.U.W.); samuele.zoratto@tuwien.ac.at (S.Z.); guenter.allmaier@tuwien.ac.at (G.A.); 4Department of Chemistry, Indiana University, Bloomington, IN 47405, USA; kmbond@iu.edu (K.B.); nalyktey@iu.edu (N.L.); mfj@indiana.edu (M.F.J.); 5Mass Spectrometric Proteomics Group, Institute of Clinical Chemistry and Laboratory Medicine, University Medical Center Hamburg-Eppendorf, 20246 Hamburg, Germany; c.krisp@uke.de (C.K.); hschluet@uke.de (H.S.); 6Partner Site Hamburg-Lübeck-Borstel-Riems, Bernhard Nocht Institute for Tropical Medicine and German Center for Infection Research (DZIF), 20359 Hamburg, Germany; munoz-fontela@bnitm.de

**Keywords:** norovirus, capsid assembly, native mass spectrometry, nES GEMMA, differential mobility analysis, CDMS

## Abstract

Noroviruses cause immense sporadic gastroenteritis outbreaks worldwide. Emerging genotypes, which are divided based on the sequence of the major capsid protein VP1, further enhance this public threat. Self-assembling properties of the human norovirus major capsid protein VP1 are crucial for using virus-like particles (VLPs) for vaccine development. However, there is no vaccine available yet. Here, VLPs from different variants produced in insect cells were characterized in detail using a set of biophysical and structural tools. We used native mass spectrometry, gas-phase electrophoretic mobility molecular analysis, and proteomics to get clear insights into particle size, structure, and composition, as well as stability. Generally, noroviruses have been known to form mainly *T* = 3 particles. Importantly, we identified a major truncation in the capsid proteins as a likely cause for the formation of *T* = 1 particles. For vaccine development, particle production needs to be a reproducible, reliable process. Understanding the underlying processes in capsid size variation will help to produce particles of a defined capsid size presenting antigens consistent with intact virions. Next to vaccine production itself, this would be immensely beneficial for bio-/nano-technological approaches using viral particles as carriers or triggers for immunological reactions.

## 1. Introduction

A vast number of nonbacterial gastroenteritis cases worldwide is caused by human noroviruses (hNoVs) [1]. Norovirus infection especially poses an acute threat to children, immunocompromised individuals and elderly people. Already a small number of particles is sufficient for infection [2]. Gastroenteritis outbreaks happen worldwide with new hNoV-variants occurring sporadically.

Human noroviruses are non-enveloped and a member of the *Caliciviridae* family. They have a positive sense, single strand, approx. 7.7 kb RNA genome organized into three open reading frames (ORFs) and a poly(A) tail. ORF1 encodes non-structural proteins, ORF2 the major capsid protein VP1, and ORF 3 the minor structural protein VP2 [3,4]. Based on VP1, noroviruses can be classified into up to ten genogroups (GI-GX) and further into genotypes [5]. Genogroups I, II, IV, VIII, and IX infect humans. The prototypical GI.1 Norwalk was isolated from stool samples in Norwalk, Ohio in 1968 [6]. Today, mostly GII.4 and GII.17 strains have been identified as a cause of viral gastroenteritis outbreaks [7,8].

There is no norovirus vaccine available yet. The lack of a robust cell culture system and small animal models as well as the immense genetic diversity of hNoVs have hindered its development to date. Although breakthroughs in developing a cell-culture system have been made in 2016 by Ettayebi et al. [9], hNoV research has mostly been based on virus-like particles (VLPs). Current vaccine candidates are also using VLPs, mostly GI.1 and GII.4 VLPs. hNoVLPs can be produced by expressing VP1 in various systems, including insect cells, yeast, mammalian cells, and plants [10,11,12,13].

Generally, VP1 can be divided into two functionally and structurally distinct domains. The shell (S)-domain, forming a scaffold around the genome, and a protruding (P)-domain. In GI.1 Norwalk, the N-terminal 225 amino acids (aa) belong to the S-domain. The P-domain is further divided into subdomains P1 and an insertion P2, with P2 being most variable and involved in host-attachment and immunogenicity [14,15]. Self-assembling properties of VP1 allow for next to fully formed *T* = 3 particles, particles of several other forms [16]. The isolated P-domain expressed in *Escherichia coli* with or without a tag can form P-dimers as well as 12-mer and 24-mer P-particles [17]. Expression of the S-domain in the baculovirus-expression system results in thin-layered, small, and smooth *T* = 3 particles [18]. In full-length VP1 particles, S- and P-domains are connected via a flexible hinge region [14]. When expressed in eukaryotic systems, caliciviruses generally are known to assemble into VP1 180-mers with *T* = 3 icosahedral symmetry. However, VP1 60-mers of *T* = 1 symmetry have been described as byproducts of hNoVLP production coexisting with other particle sizes and independent of the expression system [19]. Recently, VP1 240-mers of *T* = 4 symmetry have also been described so far only for GII.4 variants expressed in insect cells as well as in plants [20,21]. In studies on virions of different norovirus variants *T* = 3 as well as *T* = 1, formations were detected [22].

Thus, hNoVLP particle sizes are polymorphic and dynamic. Native mass spectrometry (MS) is a perfect biophysical tool to characterize these structural dynamics [23]. Previously, VLPs of three different norovirus variants have been investigated with native MS [24,25]. In our previous work, we established the pH stability pattern of two norovirus variants, GI.1 West Chester and GII.17 Kawasaki [25]. Stability was assessed in different ionic strengths as well as pH levels and compared to results on Norwalk VLPs [24]. In all three variants, *T* = 3 particles were identified as the major population. Furthermore, GII.17 Kawasaki was resistant to changing conditions, while both GI variants disassembled upon alkaline treatment.

In order to characterize hNoVLPs in detail and gain more insights into size determination, we extended our previous native MS studies with a set of biophysical methods. Next to charge detection mass spectrometry (CDMS) for mass determination of heterogeneous particle populations and proteomics, we used nano electrospray gas-phase electrophoretic mobility molecular analysis (nES GEMMA) [26], especially suited to measure high-mass particles at low concentrations [27,28]. Notably, this fast technique allows for measurements at low ionic strength and with less concentrated sample.

We described particle preparations from insect cells with sample batches, where only *T* = 1 particles were detected. A major VP1 truncation was identified in all particle preparations forming these *T* = 1 particles. We found that this size-limitation was genogroup- and genotype-independent and could not be rescued in different buffer conditions. This provides great implications for vaccine design and other applications of bio-nanoparticles, where size-homogeneity is highly favored.

## 2. Materials and Methods

### 2.1. VLP Production and Preparation

Full-length VP1 genes for GI.1 West Chester, GII.4 Saga 2006, GII.10 Vietnam, GII.17 Kawasaki308, and GII.17 Saitama T87 (GenBank accession numbers: AY502016.1, AB447457.1, AF504671.2, LC037415.1, AII73747.1) were cloned and expressed in a baculovirus system [29,30]. After transfection of a bacmid containing the recombinant VP1 gene in Sf9 insect cells and incubation for 5–7 days, the culture medium was collected and centrifuged for 10 min at 3000 rpm at 4 °C. Subsequently, Hi5 insect cells were infected with recovered baculovirus and incubated for 5 days. After centrifuging the culture medium for 10 min at 3000 rpm at 4 °C and then 1 h at 6500 rpm at 4 °C, VLPs in the supernatant were concentrated by ultracentrifugation at 35,000 rpm (Beckman Ti45 rotor, Krefeld, Germany) for 2 h at 4 °C. Furthermore, VLPs were further purified using CsCl equilibrium gradient ultracentrifugation at 35,000 rpm (Beckman SW56 rotor, Krefeld, Germany) for 18 h at 4 °C. VLPs were pelleted for 2 h at 40,000 rpm (Beckman TLA55 rotor, Krefeld, Germany) at 4 °C and solved in PBS (pH 7.4).

### 2.2. VP1 Mapping

Trypsin digestion. For tryptic digestion followed by proteomic analysis, VLP samples in PBS at 15 µM VP1 were separated via sodium dodecyl sulfate polyacrylamide gel electrophoresis (SDS-PAGE) following the reported method [31]. After staining with a solution containing 0.5% Coomassie brilliant blue R250, 50% ethanol, and 7% acetic acid, respective gel bands were cut into small pieces and further processed according to Shevchenko et al. [32]. After digestion, the samples were dried and thereafter dissolved in 0.1% formic acid and transferred into the autosampler. Tryptic peptides were either separated on a nano-UPLC system (Dionex Ultimate 3000 UPLC system, Thermo Fisher Scientific, Bremen, Germany) with a 50 cm C18 analytical column (Acclaim PepMap 100, 75, 3 µm, Thermo Fisher Scientific, Darmstadt, Germany) or a nano-UPLC system (nanoAcquity, Waters, Manchester, Great Britain) with a 25 cm C18 analytical column (BEH C18 Column, 75, 1.7 µm, 100 Å, Waters) using a 60 min gradient with increasing acetonitrile concentration from 2% to 30%. Eluting peptides were desorbed and ionized with an electrospray ionization (ESI) source into a Tribrid mass spectrometer consisting of a quadrupole, linear ion-trap, and an Orbitrap (Fusion; Thermo Fisher Scientific, Bremen, Germany) with the Dionex setup or a quadrupole Orbitrap mass spectrometer (QExactive; Thermo Fisher Scientific, Bremen, Germany) with nanoAcquity setup operated in data-dependent acquisition (DDA) mode. MS/MS spectra were searched with the Sequest algorithm integrated in the Proteome Discoverer software version 2.0, against AY502016.1, AB447457.1, AF504671.2, LC037415.1, AII73747, and a common contaminant protein database. Precursor ion mass tolerance was set to 10 ppm, and fragment ion mass tolerances was set to 0.02 (QExactive) or 0.6 Da (Fusion). Carbamidomethylation was set as a fixed modification on cysteine residues. Acetylation of the protein N-terminus, N-terminal methionine loss, the oxidation of methionine, deamidation of asparagine and glutamine, and glutamine to pyroglutamate on the peptide N-terminus were set as variable modifications. Only peptides with a high confidence (FDR of <1%) using a decoy database approach were accepted as identified.

Pepsin digestion. VLP samples were mixed 1:1 with denaturing buffer (300 mM phosphate buffer, pH 2.3, 6 M urea). Pepsin digestion of 100 pmol VP1 was performed online (Agilent Infinity 1260, Agilent Technologies, Waldbronn, Germany) on a home-packed pepsin column (IDEX guard column with an internal volume of 60 µL, Porozyme immobilized pepsin beads, Thermo Scientific, Darmstadt, Germany) at a flow rate of 75 µL/min (0.4% formic acid in water). Peptides were trapped in a trap column (OPTI-TRAP for peptides, Optimize Technologies, Oregon City, OR, USA.) and separated on a reversed-phase analytical column (PLRP-S for Biomolecules, Agilent Technologies) using a 27 min gradient of 8–40% organic solvent (0.4% formic acid in acetonitrile) at 150 µL/min. MS was performed using an Orbitrap Fusion Tribrid in positive ESI data-dependent MS/MS acquisition mode (Orbitrap resolution 120,000, 1 microscan, HCD 30 with dynamic exclusion). Precursor and fragment ions were searched and matched against a local protein database containing the proteins of interest in MaxQuant (version 1.5.7.0, Max-Planck-Institute, Munich, Germany) using the Andromeda search engine. The digestion mode was set to “unspecific” and N-terminal acetylation, deamidation, oxidation, and disulfide bond formation were included as variable modifications with a maximum number of 5 modifications per peptide. Peptides between 5 and 30 amino acids length were accepted. The MaxQuant default mass tolerances for precursor (4.5 ppm) and fragment (20 ppm) ions defined for the Orbitrap instrument (Thermo Fisher Scientific, Bremen, Germany) were used for data search. The minimum score for successful identifications was set to 20 for unmodified and 40 for modified peptides.

The mass spectrometry proteomics data have been deposited to the ProteomeXchange Consortium via the PRIDE [33] partner repository with the dataset identifier PXD023182.

### 2.3. Sample Preparation

For mass spectrometry as well as nES GEMMA analysis, hNoVLP sample solutions were exchanged to 40 and 250 mM ammonium acetate solutions. Solution pH was adjusted between 5 and 9 using acetic acid and ammonia. For the solution exchange, Vivaspin 500 centrifugal concentrators (10,000 MWCO, Sartorius, Göttingen, Germany) or Zeba micro spin^TM^ desalting columns 0.5 mL (7000 MWCO, Thermo Fisher Scientific, Rockford, IL, USA) were used. Generally, 5 filtration steps using spin filters and 3 steps using size-exclusion columns were employed. Samples were diluted to 10 µM VP1 protein or further diluted, if necessary, to obtain spectra.

### 2.4. Mass Spectrometry

Conventional native MS measurements of VLPs were performed using a quadrupole time-of-flight (QToF) instrument Q-Tof 2 (Waters, Manchester, UK and MS Vision, Almere, the Netherlands) modified for high mass experiments [34]. For ESI, capillaries were handmade by pulling borosilicate glass tubes (inner diameter 0.68 mm, outer diameter 1.2 mm with filament, World Precision Instruments, Sarasota, FL, USA) using a two-step program in a micropipette puller (Sutter Instruments, Novato, CA, USA) with a squared box filament (2.5 × 2.5 mm). Gold-coating of capillaries was performed using a sputter coater (Quorum Technologies., East Sussex, UK, 40 mA, 200 s, tooling factor of 2.3 and end bleed vacuum of 8 × 10^−2^ mbar argon or Safematic (Zizers, Switzerland), process pressure 5 × 10^−2^ mbar, process current 30.0 mA, coating time 100 s, 3 runs to vacuum limit 3 × 10^−2^ mbar Argon). Capillaries were opened on the sample cone of the mass spectrometer. Using a nanoESI source, ions were introduced into the vacuum at a source pressure of 10 mbar. The positive ion mode was used to record spectra. Generally, voltages of 1.45 kV and 165 V to the capillary and cone, respectively, were used and adjusted during spray-optimization. Xenon was used as a collision gas at a pressure of 1.7 × 10^−2^ mbar in order to enable better transmission of high-mass ions [35]. MS profile and repetition frequency of the pusher pulse were adjusted to high mass range. For instrument calibration, a cesium iodide spectrum was recorded the same day. Analysis was performed using MassLynx V4.1 SCN 566 (Waters, Manchester, UK) and Massign [36].

Charge detection mass spectrometry (CDMS) was performed on a home-built CDMS instrument described in detail elsewhere [37] in order to enable measurements of heterogeneous complexes in the MDa range or larger. Briefly, charge and *m*/*z* of single ions are measured simultaneously using a charge conduction cylinder and electrostatic ion trap. In contrast to conventional QToF MS, CDMS sidesteps the need for charge states to be assigned. Ions were generated using an automated nano-ESI source (Nanomate, Advion, Ithaca, NY, USA) with a capillary voltage of approximately 1.7 kV. After entering a heated metal capillary, ions are transmitted using various ion optics to a dual hemispherical deflection energy analyzer, which selects ions with energies centered on 100 eV/z. Subsequently, ions enter a modified cone trap where they oscillate back and forth in a charge detection cylinder for 100 ms. Single ion masses were binned to generate mass spectra. Mass spectra were analyzed by fitting Gaussian peaks with Origin software (OriginPro 2016).

Gas-phase electrophoresis was performed on a nES GEMMA instrument (TSI Inc, Shoreview, MN, USA) consisting of a nES aerosol generator (model 3480) including a ^210^Po α-particle source, an electrostatic classifier (model 3080) with a nano differential mobility analyzer (nDMA), and an n-butanol based ultrafine condensation particle counter (model 3025A). Briefly, particle-size determination is a function of the particles’ trajectory in the nDMA chamber. The trajectory of a size-specific particle is based on the sheath flow of particle-free ambient air and an orthogonal electric field applied. Therefore, with a constant high laminar sheath flow of air and a variable electrical field, only specific particle sizes can successfully be transported to the particle counter device for detection. For electrospraying, polyimide-coated fused silica capillaries (25 µm inner diameter, Polymicro, obtained via Optronis, Kehl, Germany) with in-house-made tips [38] were used. Settings for a stable Taylor cone at the nES tip were chosen, typically around 2 kV, resulting in approx. −375 nA current, 0.1 L/min CO_2_ (Messer, Gumpoldskirchen, Austria), and 1.0 L/min filtered, dried ambient air. Four pounds per square inch differential (psid, approx. 27.6 kPa) were applied to additionally move the sample through the capillary, and 15 L/min sheath flow filtered ambient air was used to size-separate VLPs in an electrophoretic mobility diameter (EMD) range from 2 to 65 nm. The corresponding EMD size range was scanned for 120 s. Subsequently, the applied voltage was adjusted to starting values within a 30 s timeframe. Seven datasets (raw data obtained from instrument software, MacroIMS manager v2.0.1.0) were combined via their median to yield a corresponding spectrum. Lastly, Gauss peaks were fit to spectra via Origin software (OriginPro 2016) to obtain EMD values. EMD values were correlated to particle mass using MW-correlation either based on proteins [39] or VLPs [28].

### 2.5. Electron Microscopy

For imaging using transmission electron microscopy (EM), hNoVLPs stored in PBS were adsorbed onto glow discharge-activated carbon-coated grids (Science Services, Munich, Germany). After three consecutive washing steps with distilled water, the sample coated grids were stained with 1% uranyl acetate. Image acquisition was performed using a FEI Tecnai^TM^ G2 transmission electron microscope and wide-angle Veleta CCD camera (FEI, Thermo Fisher Scientific, USA and Olympus, Tokyo, Japan) at 80 kV.

## 3. Results

### 3.1. Truncated GII.4 Saga VP1 Forms Homogeneous T = 1 Particles

Here, we extended our previous investigations on GI.1 West Chester and GII.17 Kawasaki to other hNoVLP constructs. VLPs of an outbreak strain GII.4 Saga are produced in the same baculovirus-expression system. Native mass spectra reveal the lack of *T* = 3 particles at neutral pH and moderate ionic strength (Figure 1). Notably, the identified peak distribution is almost baseline resolved, indicating a highly homogeneous population annotated to VP1 60-mers or *T* = 1 particles. An additional, unresolved peak distribution around 15,000 *m*/*z* relates to metastable ions. Metastable ions are commonly accompanying high-mass ions as these disintegrate partially in the ToF-analyzer, as such they cannot be targeted by selection in the quadrupole, allowing differentiation from ions originating from the sample solution. Figure 1 also illustrates collision-induced dissociation (CID) products for GII.4 Saga. The *T* = 1 ions (~150+ charges) dissociate via consecutive losses of VP1 monomers, with at least two subspecies in mass, as well as corresponding high mass ions, VP1 59-mer, 58-mer, and 57-mer. Mass-assignment of the dissociated monomer suggests an N-terminal truncation of 45 amino acids (aa) of the main species and a subpopulation lacking 45 aa. Proteomics data following trypsin-digestion as well as pepsin-digestion (Table 1) results in VP1 sequence coverage of 95 and 90% with the N-terminal coverage starting from residues 25 and 27, indicating additional subpopulations, which are low abundant. Notably, the C-terminus is complete up to several arginine residues (C-terminal three to six residues), which exclude coverage for both proteins due to small peptides. This suggests exclusive N-terminal truncation. An assembly into a 60-mer of the full-length VP1 would result in a theoretical *T* = 1 mass of 3.54 MDa, and VP1 lacking 45 aa would form 3.28 MDa *T* = 1 particles. The assigned mass of 3.27 MDa using QToF MS and 3.35 MDa using native CDMS (Appendix A; note, masses in CDMS are always higher, indicating incomplete desolvation) suggests that detected 60-mer particles are indeed formed mainly from VP1 lacking at least 45 aa. To conclude, in this case, heterologous expression of GII.4 Saga VP1 results in a truncated VP1 species with the mere ability to form *T* = 1 but not *T* = 3 particles.

### 3.2. T = 1 Capsid Formation Is Genotype-Independent

Additional hNoVLPs were investigated to pinpoint whether the truncation seen in GII.4 Saga causes *T* = 1 formation. Norovirus-like particle polymorphism has been described as putatively genotype-dependent [20,21,40]. Therefore, we extended our sampling to GII.10 Vietnam and GII.17 Saitama (Figure 1). In line with GII.4 Saga measurements, most abundant peak distributions were assigned to VP1 60-mers for both variants. Notably, more acceleration energy compared to GII.4 Saga was needed to gain charge-resolution for VP1 60-mer peaks, which indicated increased VP1 heterogeneity in these samples (Appendix A). Furthermore, *T* = 1 ions in GII.10 Vietnam showed tailing with a non-resolved shoulder peak, indicating either aggregation or a further low-intensity assembly of slightly higher mass. In GII.17 Saitama mass spectra, heterogeneity was even more prominent as multiple higher-mass assemblies gave rise to complex ion distributions between 30,000 and 40,000 *m*/*z*. GII.17 Saitama ion distributions were overlapping with the respective *T* = 3 *m*/*z* range observed in previous mass spectra, but clear mass assignment was hindered due to high heterogeneity in the sample. Dissociated VP1 monomer species for all listed variants except GII.17 Saitama, where signal intensities were too low for selective dissociation experiments and monomer mass was inferred from CDMS (Appendix A), are listed in Table 1. If a similar incomplete desolvation for GII.17 Saitama as for GII.4 Saga was assumed in CDMS, the VP1 monomer mass further reduced by ~1500 Da corresponding to an additional 14 aa missing, resulting in a total of 31 aa, closer to the values observed for the other hNoVLPs. Mutual in most VP1 monomer measurements was a major truncation of at least 45 aa (45 aa in GII.4, 45 aa in GII.10, and 17 aa/31 aa in GII.17). Although VP1 truncation was similar in all three variants, the putative cleavage site did not reside in a conserved region, and a putative protease could not be assigned (Table 1).

### 3.3. Heterologous Expression of GI.1 West Chester Results in Either T = 1 or T = 3 Preparations

To provide further evidence of truncation influence, we compared two GI.1 West Chester batches. Batch 1 is identical to the sample used in our previous work [25]. In the second batch, no *T* = 3 particles were detected at neutral pH using native MS (Figure 2). The main peak distribution was assigned to *T* = 1 particles, which was accompanied by a low-intense shoulder peak comparable to GII.10 Vietnam. At increased acceleration voltage, the *T* = 1 ions released VP1 monomers. A close-up of these monomers showed that at least two subspecies were present. This directly contributed to heterogeneity and therefore low peak resolution of higher-mass species. The dominating VP1 species was assigned to 52,760 ± 10 Da, or the theoretical VP1 mass lacking 40 N-terminal aa. Proteomics data, which hinted to subspecies with minor truncations, was consistent with other variants tested in this study (Table 1). In our previous study, we could identify GI.1 West Chester VP1 monomers with the major species lacking only three amino acids, forming mainly *T* = 3 particles [25]. Taken together, we can assume that with the VP1 N-terminus of GI.1 West Chester lacking three amino acids the formation of *T* = 3 particles is possible, while with an expanded truncation of 40 amino acids this is no longer the case.

Furthermore, we characterized particle size and stoichiometry in further detail using nES GEMMA. Measurements of both GI.1 West Chester preparations are superimposed in Figure 3. In order to exclude artefacts, all samples were measured at different dilutions (Appendix A). Putative artefacts included unspecific, nES-based aggregates at high sample concentrations, as well as multiply charged particles obtained at low percentage values during charge equilibration in the bipolar atmosphere of the spray chamber. Comparison of both batches at neutral pH reveals a clear shift in particle size and their counts. In the first GI.1 West Chester batch, most prominent particle counts were at 34.37 ± 0.13 nm, which was assigned to *T* = 3 particles. Further particle counts at 8.10 ± 0.05 and 24.09 ± 0.27 nm were assigned to VP1 dimer and VP1 60-mer. nES GEMMA spectra of the second batch showed a predominant species at 24.50 ± 0.12 nm equaling *T* = 1, as well as a species with low counts at 30.71 ± 0.17 nm. The population at 30.71 nm was assigned to 6.24 MDa using VLP correlation fitting approximately 120 VP1 (Appendix A) [28]. At pH 9, high-mass particles in batch 1 were fragile, complementing our previous findings with native MS [25], while the *T* = 1 VLPs in batch 2 were resistant to pH 9. Other particles in the second batch bigger than 24.18 ± 0.06 nm disappeared. Interestingly, comparing EMDs of this 60-mer species at neutral pH and pH 9 did not indicate swelling or shrinking of the particles. Notably, the VP1 dimers released in alkaline conditions were slightly smaller in the second batch in line with the observed truncation. Moreover, size difference was not observed for *T* = 1 particles in the two batches in line with an N-terminal truncation located at the inner face of the capsids. We can conclude that no *T* = 3 particles were detected with nES GEMMA in the second West Chester batch, which indicated that at least a certain amount of full-length or less truncated VP1 subpopulation was needed to form *T* = 3 particles. Intermediate-sized populations could stem from either truncated, full-length VP1, or a mixture.

### 3.4. Detailed nES GEMMA and CDMS Profiling

As nES GEMMA is fast and sensitive, all samples were further profiled to see if *T* = 3 assemblies could be rescued and/or *T* = 1 particles from truncated VP1 were in general more stable at alkaline pH. At neutral pH and low ionic strength, particle size patterns of all variants were in line with conventional native MS. Next to the GI.1 West Chester second batch, GII.4 Saga and GII. 10 Vietnam formed *T* = 1 particles but not *T* = 3 particles. GII.17 Saitama showed some signals, which may have originated from *T* = 3 particles. Similar to the low-count species in GI.1 West Chester batch 2 of 30.71 ± 0.17 nm (120-mer), in GII.4 Saga and even more prominent in GII.10 Vietnam, further particles were detected at around 33 nm, equaling 7.7 MDa (VP1 140-mer). In line with native MS data, enormous heterogeneity was observed in GII.17 Saitama, and multiple species other than VP1 60-mer could be distinguished with nES GEMMA (Figure 4). Measurements at pH 5 up to pH 9 revealed that *T* = 1 formations of all samples were mostly resistant to changing solution conditions (Figure 5). Starting from pH 8, free VP1 dimer was detected in all variants in low counts. At pH 9, GII.10 Vietnam and GII.17 Saitama showed reversed particle count ratios of VP1 60-mer and VP1 dimer and larger assemblies did not disintegrate. In GII.4 Saga, no complexes were detected at pH 9 and the employed low VP1 concentrations. However, GII.4 Saga 60-mers remained intact at alkaline pH and low ionic strength in conventional native MS measurements at 10 µM VP1 (50 mM ammonium acetate, Appendix A). For all samples, particle-size patterns were also consistent at pH 5, although with lower particle counts and with increased background noise. Taken together, this indicated that *T* = 1 particles were highly stable, resisting alkaline pH, and *T* = 3 particle formation could not be rescued by changing solution conditions.

So far, several different size-populations have been detected outside the scope of the Caspar–Klug capsid assembly theory [41], where multiples of 60 (with 120 being formally not allowed) form particles of icosahedral symmetry. CDMS measurements in conditions comparable to our conventional QToF measurements at 250 mM ammonium acetate were used to unambiguously assign such assemblies (Figure 6). For GII.4 Saga, no species but *T* = 1 particles were observed in sufficient counts to fit peaks. Albeit peaks at approximately 33 nm appeared in low-salt nES GEMMA measurements, no respective peak could be assigned to VP1 140-mer with CDMS. At notably lower ion counts, GII.10 Vietnam showed *T* = 1 particles, as well as two further species with approximately 4.5 and 6.9 MDa assigned to VP1 79-mer and VP1 121-mer, respectively. Note that those species were approximations due to very low counts, and stoichiometry was based on assuming that 60 VP1 formed the 3.41 MDa population. For both variants, the VP1 mass inferred from the 60-mer was higher than determined in conventional native MS (CDMS/nMS: GII.4 Saga 55.8/54.6 and GII.10 Vietnam 56.8/55.6 kDa). This indicated mixed subpopulations of different VP1-size forming particles or less efficient desolvation in CDMS compared to QToF measurements. Notably, the species at 4.5 MDa was repetitive in GII.10 Vietnam, as well as GI.1 West Chester batch 1 (Appendix A). In GII.17 Saitama, CDMS helped to elucidate mass heterogeneity observed in the other methods. Next to a distinct population of 3.44 MDa assigned to VP1 60-mer, five additional high mass species could be deconvoluted. CDMS clearly showed the absence of fully-formed *T* = 3 particles. Proteomics data for Saitama indicated a subpopulation of VP1 with a minor truncation of 3 aa. This subpopulation would putatively be able to form *T* = 3 particles in low amounts, which would likely be prone to disassemble at varying concentrations, ionic strengths, and pH levels. Given a mass of approximately 57.3 kDa, CDMS Saitama high-mass species fit VP1 71-, 91-, 100-, 108-, and 120-mers. In contrast, the stability at alkaline pH and low concentrations of these species suggested distinct assemblies. Particle size-heterogeneity was also observed using EM (Appendix A), where various larger assemblies were detected in GII.17 Saitama micrographs. Moreover, although formally not allowed according to triangulation theory, observed VP1 intermediates were repeatedly found in all tested variants and with different techniques.

## 4. Discussion

In this study, different hNoVLP variants were investigated with a set of biophysical tools in order to obtain insights into particle size, stoichiometry, and shape. hNoVLP preparations forming *T* = 1 particles were identified. *T* = 1 particles were repeatedly described in hNoVLP preparations [19,21,24,25]. In our previous study, *T* = 1 and *T* = 3 particles were coexisting in GI.1 West Chester and GII.17 Kawasaki preparations at neutral pH. In line with a former study on GI.1 Norwalk [24], GI.1 West Chester formed *T* = 3 particles, which were prone to disassemble in alkaline pH [25]. Here, we identified a major VP1 N-terminal truncation of more than 40 aa in several hNoV variants, leading to *T* = 1 particles only. The origin of this truncation was unclear. As no clear conserved cleavage motif could be identified, various or unspecific proteases were proposed, likely originating from the insect cell expression system [42,43]. Moreover, this pointed at a structurally defined proteolytic site, which was in line with the flexibility observed in the N-terminal arm. Notably, in all investigated preparations, less populated subspecies with limited truncations building the observed particle formations could not be excluded as proteomics data suggest. This was further supported by a *T* = 1 structure from cryo-EM [21], which showed no electron density for the N-terminal stretch, indicating that it was either flexible or absent in the preparation.

The ability of truncated VP1 to form mainly *T* = 1 particles was genogroup- and genotype-independent, as several hNoV variants were targeted here. This indicated a major truncation was sufficient for VP1 to form *T* = 1 only, and therefore, homogenous, small-sized hNoVLP production was reproducible. Interestingly, several intermediate species were observed repeatedly. In GI.1 West Chester batch 2, GII.4 Saga, and GII.10 Vietnam intermediates were detected in very low proportions. In GII.17 Saitama, spectra suggested a heterogeneous size distribution of several high-mass species with increased counts. A repetitive species that overlapped between preparations was VP1 120-mer. VP1 dimer has been described as a building block for capsid assembly, which suggests that intermediate species must be even integers [14,15,40,44]. Therefore, GII.17 Saitama CDMS measurements resulting in odd numbered complexes were rounded here. Using a combination of characterizing tools like nES GEMMA, conventional MS, and CDMS, VP1 120-mer appeared biologically relevant, although not allowed according to triangulation theory [41]. Moreover, the agreement in GEMMA mass assignment based on a VLP calibration and CDMS revealed that these assemblies resemble hollow spheres like regular capsids. Putative, non-allowed *T* = 2 particles were described for bluetongue virus and brome mosaic virus among others [45,46,47]. Another intermediate observed in different hNoVLP preparations was VP1 80-mer. Interestingly, it was detected in preparations, in which *T* = 3 particles were also observed like GI.1 Norwalk [24], GI.1 West Chester [25] (Appendix A). This indicated different behavior of full-length and truncated VP1 and mixtures thereof. General observed particle plasticity suggested that these species could be trapped formations or overgrown particles, as observed for hepatitis B virus and woodchuck hepatitis virus [48,49]. However, it has to be noted that assemblies have specific sizes as evident from CDMS, rather than covering a broad distribution. The inability to form full *T* = 3 particles indicated that the N-terminus was required to form flat C/C dimers, leading to lattice expansion. We already proposed an influence of the N-terminus in capsid size determination in our previous study [25]. In turn, the truncated VP1 would likely form mainly bent A/B-like dimers, forming the pentameric vertices present in both VLP formations. This would then likely preclude *T* = 2 capsids. An alternative assembly route would follow octahedral symmetry, which has been described for SV40 polyomavirus [50]. This requires strongly bent dimer interactions and a 24-mer octahedron formed from pentamers exactly matching the VP1 120-mers detected here. This interpretation was further appealing as it offered an explanation for the aberrant GII.17 Saitama assemblies of 70/72, 90/92, 100, and 108/110 VP1 subunits being octahedrons lacking multiple pentamers. Polymorphism in hNoVLP production, independent of the expression system, have been described both for VP1 forming *T* = 1 and *T* = 3 particles at neutral pH levels and intermediates upon changing conditions. In an assembly study on GI.1 Norwalk, three N-terminal deletion mutants were compared [18]. Full-length as well as deletion of N-terminal 20 aa still resulted in *T* = 3 particles. Deletion of 34 and 98 aa N-terminally did not result in any particles detectable with electron microscopy. However, N-terminal 34 aa mutant expression was described as low, hampering the assessment of how this deletion is involved in capsid assembly. Furthermore, N-terminal deletions of 26 and 38 aa were introduced in GII.4 Sydney VLPs. Both constructs were found to form mainly small particles when examined with electron microscopy [51]. Next to deletion itself, culture conditions were also described as a putative reason for size heterogeneity. Another attempt to gain size-homogeneity in hNoVLP preparations was performed by Someya et al. using GI.4 Chiba VLPs [52]. Truncation of 45 aa N-terminally, similar as observed in this study, was identified and the subsequent introduction of a mutation Leu43Val in this region resulted in the formation of merely *T* = 3 particles. However, in a follow-up study, GI.4 Chiba mutants were shown to form 23 nm or *T* = 1 particles, putatively due to freezing and thawing of preparations or pH-dependent processes [53]. Previously, *T* = 4 particles were identified in hNoVLP preparations [20,21]. Interestingly, one study included GII.2 Snow Mountain virus forming *T* = 1 particles [21]. Here, residues 1 to 46 were not covered in electron density maps. Hence, truncation as the origin of small particles similar to our observations could not be excluded.

Next to particle size distribution, the influence of solution pH was investigated. *T* = 1 particles, as well as higher-mass assemblies in GII.17 Saitama, were found to be pH-independent. Moreover, in preparations forming mainly *T* = 1 particles, like GI.1 West Chester batch 2, *T* = 1 particles showed increased stability in alkaline conditions. Therefore, truncated VP1 was able to build particles with increased stability. This implies great advances for bio-nanotechnology, as especially in approaches using VLPs as carrier particles, they need to be stable independently of environmental conditions. The contribution of the N-terminus to pH stability suggests a way to obtain S-particles of increased stability by truncation.

## 5. Conclusions

There is no hNoV vaccine available yet and hNoVLP size polymorphism could contribute to this circumstance. Therefore, N-terminally truncated particles have great potential to be beneficial as they imply size homogeneity. An N-terminal truncation of VP1 also leaves P-domains and therefore protrusions on assembled particles intact, as evident from EM data on all tested variants. Studies on P-particles imply the necessity of protrusions for antigen recognition and immunogenicity [54]. Furthermore, P-particles were shown to putatively enable other immunological approaches like antigen presentation [55]. There are clear structural differences between *T =* 1 particles of truncated VP1 and P-particles as the S-domain is missing in the latter. Furthermore, the S-domain is generally more conserved among hNoVs, putatively allowing induction of cross-reactive antibodies. Orientation of dimeric protrusions, and therefore their interaction is likely to be different in P-particles missing the S-domain, truncated VP1 *T =* 1 or full-length *T* = 3 particles. Whether this affects antibody raising and therefore immunogenic reaction needs to be investigated. Additionally, increased stability would likely allow for simplified and prolonged storage. Our results indicate that such small particles from truncated VP1 can be produced independent of genotype by introducing N-terminal deletion mutants.

## Figures and Tables

**Figure 1 vaccines-09-00008-f001:**
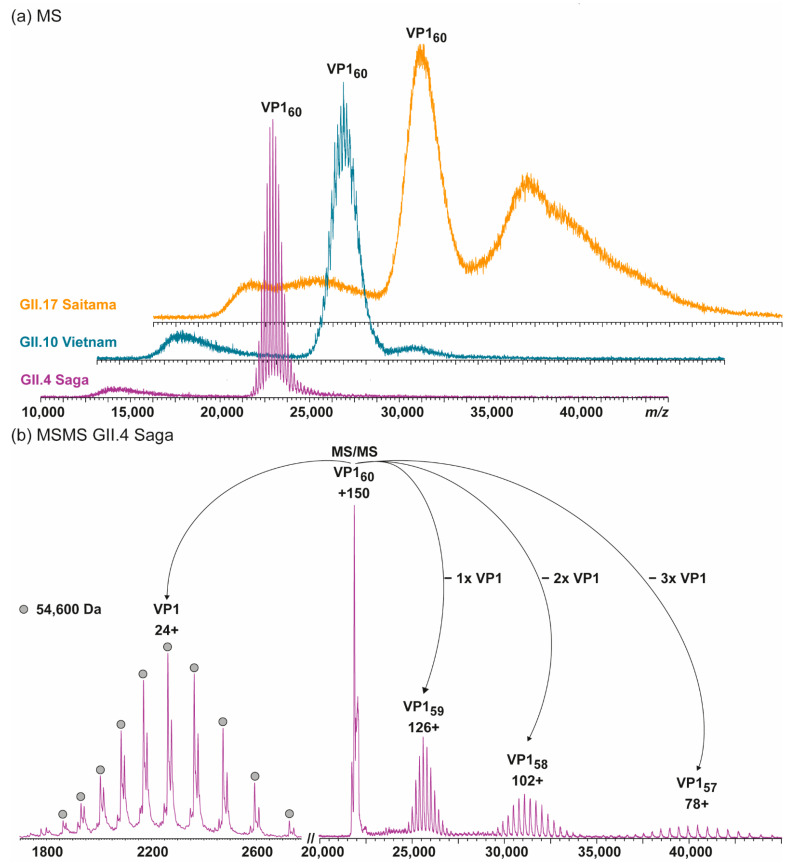
Native MS of different human norovirus-like particles (hNoVLPs) suggests that a major truncation of VP1 leads to *T* = 1 particles. (**a**) From bottom to top spectra of GII.4 Saga (purple), GII.10 Vietnam (blue), and GII.17 Saitama (orange) in 250 mM ammonium acetate pH 7 at 10 µM VP1 are shown. All variants have main ion distributions between 20,000 and 25,000 *m/z*, which are assigned to VP1_60_ complexes. GII.10 Vietnam and especially GII.17 Saitama also form larger assemblies as indicated by additional signal above 25,000 *m*/*z*. (**b**) Collision-induced dissociation MS/MS is shown exemplarily for GII.4 Saga. The dissociation of the 150+ charged VP1 60-mer into VP1-monomer (top left) and residual VP1 59-, 58-, and 57-mer (top right) is shown. Charge states and average monomer mass are annotated. The MS/MS confirms stoichiometry assignment and reveals monomer truncation.

**Figure 2 vaccines-09-00008-f002:**
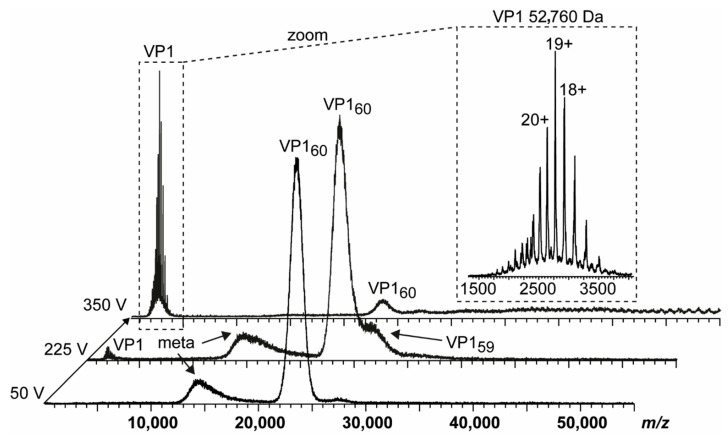
Native MS of a GI.1 West Chester batch forming merely VP1_60_ complexes. Dissociation pathway without selection in the quadrupole is shown for GI.1 West Chester in 250 mM ammonium acetate pH 7 at 10 µM VP1. From bottom to top, illustrative mass spectra are shown for 50, 225, and 350 V acceleration into the collision cell. While at 225 V, VP1 monomers dissociate with the main population of VP1 60-mer still intact, the signal ratio of VP1 monomer:60-mer is reversed at 350 V. An insert shows a zoom of dissociated VP1 monomer with annotated charge states and average mass. As lower mass ions at approximately 15,000 *m/z* are annotated as metastable ions (meta), monomer lacking at least 40 aa most likely dissociate from *T* = 1 species.

**Figure 3 vaccines-09-00008-f003:**
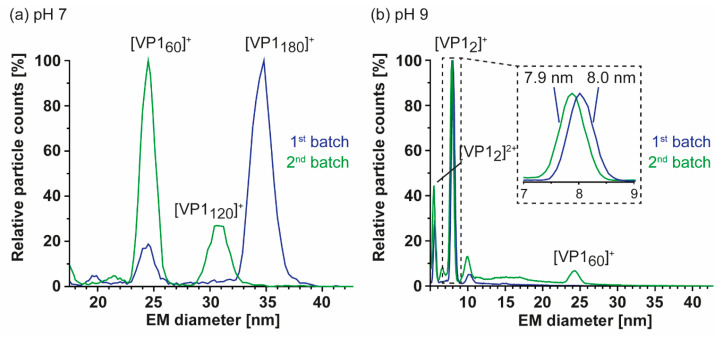
Comparison of two GI.1 West Chester VLP preparations using nano electrospray gas-phase electrophoretic mobility molecular analysis (nES GEMMA). Illustrative spectra are shown for both samples in 40 mM ammonium acetate at pH 7 (**a**) and pH 9 (**b**) at approximately 2–10 µM VP1. Depicted are exemplary spectra for two batches in blue (1) and green (2). (**a**) Batch 1 shows a clear pattern with a main population of 34.37 ± 0.13 nm and less particle counts at 24.09 ± 0.27 nm, assigned to VP1 180-mer and VP1 60-mer. In batch 2, VP1 60-mer detected at 24.50 ± 0.12 nm is the most abundant species. Furthermore, a species at 30.71 ± 0.17 nm assigned to VP1 120-mer is present and VP1 180-mer is missing. (**b**) At pH 9, no VP1 complexes other than VP1 dimer were detected for batch 1. The second batch shows a small particle distribution at 24.18 ± 0.06, indicating higher stability of VP1 60-mers. A zoom at the electrophoretic mobility diameter (EMD) range depicts a minor difference in VP1 dimer size of 8.03 ± 0.01 nm for batch 1 and 7.89 ± 0.01 nm for batch 2. For both conditions, the shown EMD range was adjusted and the complete range including multiply charged species in the low EMD range are shown in Appendix A.

**Figure 4 vaccines-09-00008-f004:**
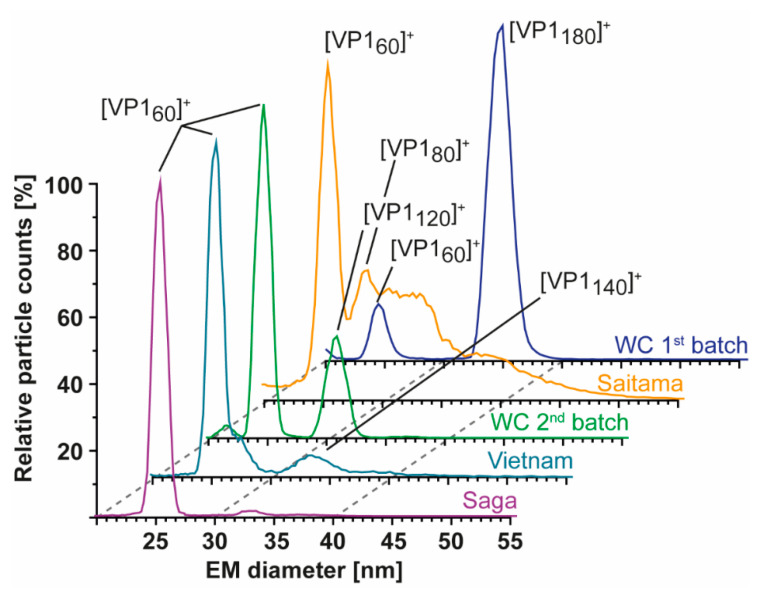
Size-distribution overview of different hNoVLPs with nES GEMMA. All variants are measured at 40 mM ammonium acetate pH 7 and approximately 2–10 µM VP1. From bottom to top GII.4 Saga (purple), GII.10 Vietnam (light blue), GI.1 West Chester batch 2 (WC, green), GII.17 Saitama (orange), and GI.1 West Chester batch 1 (WC, dark blue). West Chester batch 1 is shown as an indication of the expected EMD range for *T* = 3 particles. Assigned species are annotated. VP1 60-mers were detected in all variants, with less counts in WC batch 1. Next to 60-mers, GII.4 Saga and GII.10 Vietnam show distributions at approx. 33 nm assigned to VP1 140-mer, and WC second batch shows a distinct peak at 30 nm assigned to VP1-120mer. In GII.17 Saitama, at least two peaks can be fitted in the particle distribution accompanying VP1 60-mer annotated as VP1 80-mer and putatively VP1 180-mer.

**Figure 5 vaccines-09-00008-f005:**
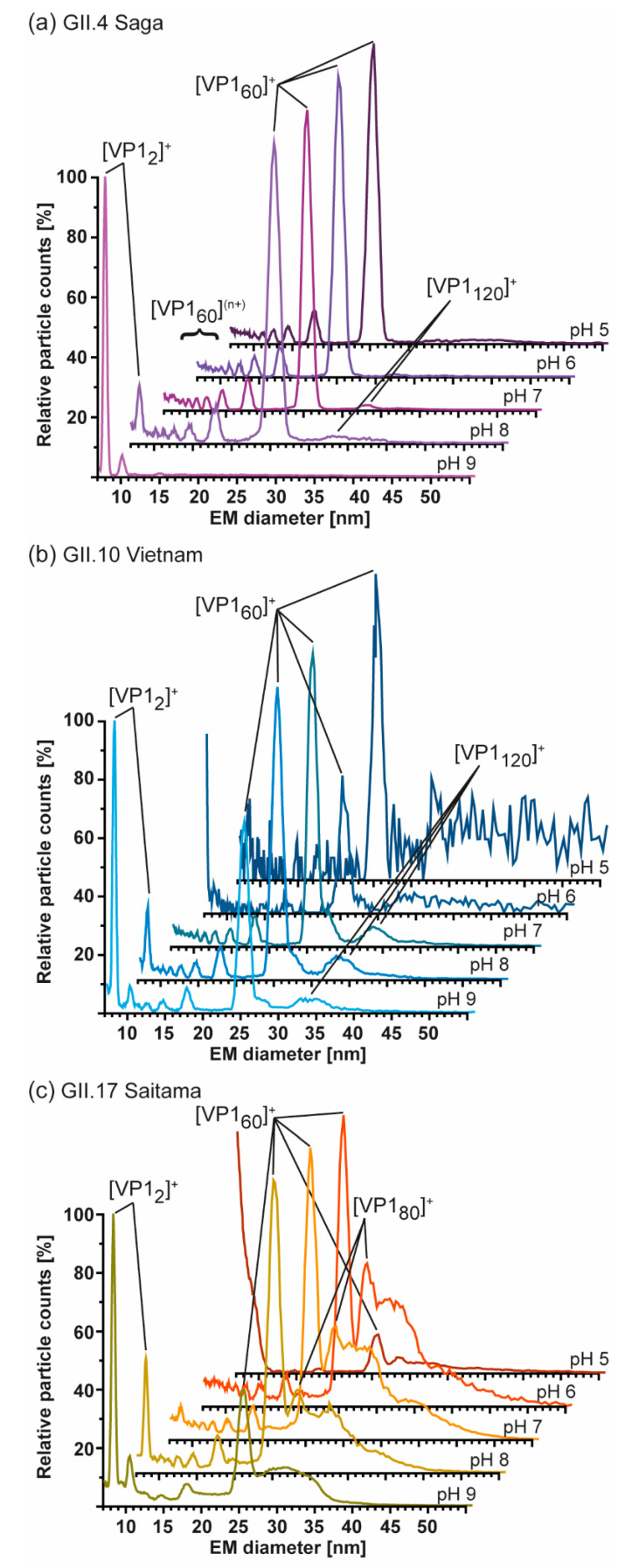
Measurements at different pH of hNoVLPs with GEMMA indicates high pH-resistance of *T* = 1 particles formed from truncated VP1. All measurements were performed at 40 mM ammonium acetate at pH 5–9 from top to bottom at approximately 2–10 µM. (**a**) GII.4 Saga (purple) shows mainly VP1 60-mer accompanied by multiply charged VP1-60-mer. Particle patterns differ only at pH 8, where VP1 dimers are present in low counts as well as pH 9, where merely VP1 dimer is detected. (**b**) GII.10 Vietnam (blue) shows intact *T* = 1 particles at all tested pH values. Increased particle counts at pH 7–9 are accompanied with multiply charged VP1 60-mer. Disassembly into VP1 dimer is seen at pH 8 but is only resulting in less VP1-mer counts at pH 9. This pattern is comparable to measurements of GII.17 Saitama (**c**). Here, main VP1-mers are accompanied by heterogeneous subspecies, which are reduced under alkaline conditions.

**Figure 6 vaccines-09-00008-f006:**
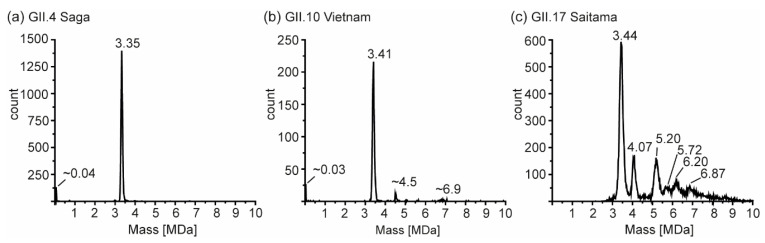
Charge detection mass spectrometry of hNoVLPs at 250 mM ammonium acetate pH 7 and 10 µM VP1. Illustrative spectra shown for (**a**) GII.4 Saga (**b**) GII.10 Vietnam, and (**c**) GII.17 Saitama. Distinct peaks are annotated; for GII.4 Saga and GII.10 Vietnam low-count species, masses are approximations.

**Table 1 vaccines-09-00008-t001:** Overview of investigated samples forming mainly *T* = 1 particles. Theoretical (th.) VP1 mass and amino acid (aa) number given for constructs West Chester, Saga, Vietnam, and Saitama constructs. Experimental (exp) mass and truncation given for main observed monomeric species after dissociation in conventional MS for all variants except GII.17 Saitama. For Saitama, mass was approximated using charge detection mass spectrometry (CDMS) (Appendix A, for nMS see Appendix A). MW: Molecular weight.

Variant	VP1 th.	VP1 exp.	Putative Cleavage Site	Trypsin Digestion	Pepsin Digestion
	MW, Total	Main Species MW Truncation	According to exp. VP1 MW	Sequence Coverage %, Minimal N-terminal Truncation	
GI.1 West Chester	56,609 Da, 530 aa	52,760 Da, −40 aa	LAMDPVAGSS/TAVATAGQVN	80%, −6 aa	98%, −2 aa
GII.4 Saga	59,005 Da 540 aa	54,600 Da, −45 aa	AIAAPVAGQQ/NVIDPWIRNN	95%, −25 aa	90%, −27 aa
GII.10 Vietnam	59,901 Da 548 aa	55,560 Da, −aa	SLAAPVTGQT/NIIDPWIRMN	95%, −27 aa	94%, −27 aa
GII.17 Saitama	58,957 Da 540 aa	57,300 Da, −17 aa	SNDGATGLVP/EINNETLPLE	91%, −32 aa	99%, −3 aa

## Data Availability

The mass spectrometry proteomics data have been deposited to the Proteo-meXchange Consortium via the PRIDE [33] partner repository with the dataset identifier PXD023182. All other data are available on request from the corresponding author.

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
