# Peer review of "N-terminal VP1 Truncations Favor T = 1 Norovirus-Like Particles"

_vaccines, 2020, doi:10.3390/vaccines9010008_

Round 1
Reviewer 1 Report
In the manuscript by Pogan et al, the authors use high resolution mass spectrometry to analyze norovirus particles produced in insect cells in order to determine the approximate size of the particle produced and its symmetry. The data presented suggests a non-specific truncation which occurs on the N-terminus of the VP1 protein promotes the formation of T=1 particles, whereas norovirus is normally considered T=3 (though T=1 particles have been described in other publications). The T=1 particle appears more stable and thus could be considered a good vaccine candidate, especially as many of the immunogenic variants are likely to be present. The also potential detect T=2 particles, which as they point out, should not occur. However, the explanations offered in the discussion are convincing. I have only minor concerns with the writing of the manuscript and presentation of some of the data. A detailed proof reading will be necessary prior to publications as I may have missed some errors.
- I am unsure what the zoom image is supposed to be. This needs a better explanation.
- line 24, VP1 needs to be defined the first time it appears in the text
- Line 63: Self-assembling properties of VP1 allow next to fully formed, virion resembling particles several other forms. Other forms of what? This is confusing
- Line 74 polymorph... should this be polymorphic?
- Line 302... characterize should be characterized
- Line 339 Wester shoudl be West
- Line 348 - Interestingly, also the larger assemblies are still observable. This is a clunky phrase and needs to be re-written
- Line 390 - I am not sure what detected is referring too?
- Line 399- merely seems to be a poor word choice
- Line 474 - antibody raising? Not sure what the authors are referring to here, but it needs correcting.
- The authors claim: Furthermore, these particles have intact protrusions, which are necessary for antigen recognition and putatively enable other immunological approaches like antigen presentation. Not sure what approaches means here, but the authors also offer no proof of this statement. Are their references to back up this claim? If not, the authors may wish re-phrase this sentence
Author Response
In the manuscript by Pogan et al, the authors use high resolution mass spectrometry to analyze norovirus particles produced in insect cells in order to determine the approximate size of the particle produced and its symmetry. The data presented suggests a non-specific truncation which occurs on the N-terminus of the VP1 protein promotes the formation of T=1 particles, whereas norovirus is normally considered T=3 (though T=1 particles have been described in other publications). The T=1 particle appears more stable and thus could be considered a good vaccine candidate, especially as many of the immunogenic variants are likely to be present. The also potential detect T=2 particles, which as they point out, should not occur. However, the explanations offered in the discussion are convincing. I have only minor concerns with the writing of the manuscript and presentation of some of the data. A detailed proof reading will be necessary prior to publications as I may have missed some errors.
I am unsure what the zoom image is supposed to be. This needs a better explanation.
We thank the reviewer for this comment. We have changed Figure 1, Figure 2 and Figure S1 to clarify. In Figure 1, the order of spectra was changed (MS spectra above, MSMS example for GII.4 Saga below) and a description (a, b) was inserted in order to ensure a clear assignment of different spectra according to figure description. Furthermore, low m/z as well as high m/z range for the inserted MSMS spectrum of GII.4 Saga (Figure 1) and the inserted MS spectrum for GII.10 Vietnam (Figure S1) were aligned on one level. In Figure 2, the arrow indicating dissociation was deleted as this might have been misleading and the zoom-in on the VP1 monomer peak was clearly marked using a line connecting the two zoom-boxes.
line 24, VP1 needs to be defined the first time it appears in the text.
This is correct and was changed by inserting the description major capsid protein (line 24).
Line 63: Self-assembling properties of VP1 allow next to fully formed, virion-resembling particles several other form. Other forms of what? This is confusing
We agree that this sentence is confusing and rephrased lines 63 to 65.
Line 74 polymorph... should this be polymorphic?
Line 302... characterize should be characterized
Line 339 Wester shoudl be West
All fixed.
Line 348 - Interestingly, also the larger assemblies are still observable. This is a clunky phrase and needs to be re -written
We thank the referee for this comment and have changed the sentence (line 365).
Line 390 - I am not sure what detected is referring too?
We agree that this phrasing is confusing and exchanged detected to observed (now line 409) as “detected intermediates” is referring to all species that have been observed with different methods in this study.
Line 399- merely seems to be a poor word choice delete
We agree and deleted the word merely (line 418).
Line 474 - antibody raising? Not sure what the authors are referring to here, but it needs correcting.
See below.
The authors claim: Furthermore, these particles have intact protrusions, which are necessary for antigen recognition and putatively enable other immunological approaches like antigen presentation. Not sure what approaches means here, but the authors also offer no proof of this statement. Are their references to back up this claim? If not, the authors may wish re-phrase this sentence
We have rewritten the discussion to clarify the impact on immunity and vaccine production including references, which hopefully clarifies these aspects (lines 490-500).
Or please see attachment.

Reviewer 2 Report
The paper I reviewed “N-terminal VP1 truncations favor T=1 norovirus-like particles” represent an extension, by using besides native mass spectrometry, gas-phase electrophoretic mobility molecular analysis and proteomics, of previous studies of the Authors in which they established the pH stability pattern of norovirus GI.1 West Chester and GII.17 Kawasaki in different ionic strength as well as pH. In particular in this study, the Authors investigated employing these biophysical tools different human Norovirus virus-like particles variants (GI.1 West Chester, GII.4 Saga 2006, GII.10 Vietnam, GII.17 Kawasaki308 and GII.17 Saitama T87 ) to obtain insights into particle size, stoichiometry and shape. While noroviruses usually form mainly T=3 particles, the Authors identified a major VP1 N-terminal truncation of more than 40 aa in several human NoV variants leading to T=1 particles only.
The article is well written, interesting, and innovative. The statements described are supported by detailed presented data. I have only two observations.
1)Did the Authors observed the VLPs by EM?
2)Although the discussion section was interesting and meaningful, I suggest the Authors to discuss more in depth the implications of the findings of their studies for vaccine design.
Author Response
The paper I reviewed “N-terminal VP1 truncations favor T=1 norovirus-like particles” represent an extension, by using besides native mass spectrometry, gas-phase electrophoretic mobility molecular analysis and proteomics, of previous studies of the Authors in which they established the pH stability pattern of norovirus GI.1 West Chester and GII.17 Kawasaki in different ionic strength as well as pH. In particular in this study, the Authors investigated employing these biophysical tools different human Norovirus virus-like particles variants (GI.1 West Chester, GII.4 Saga 2006, GII.10 Vietnam, GII.17 Kawasaki308 and GII.17 Saitama T87 ) to obtain insights into particle size, stoichiometry and shape. While noroviruses usually form mainly T=3 particles, the Authors identified a major VP1 N-terminal truncation of more than 40 aa in several human NoV variants leading to T=1 particles only.The article is well written, interesting, and innovative. The statements described are supported by detailed presented data. I have only two observations.
1)Did the Authors observed the VLPs by EM?
Such data is indeed useful for comparison. Negative stain EM has now been performed on all samples, the data has been implemented in the supplement (Figure S5) and is referred to in the main text along the data of Figure 6 (lines 407-408).
2)Although the discussion section was interesting and meaningful, I suggest the Authors to discuss more in depth the implications of the findings of their studies for vaccine design.
We thank the reviewer for the suggestion and provide now a more profound discussion on these implications (lines 490-500).
Or please see attachment.
